# Seeking the Amygdala: Novel Use of Diffusion Tensor Imaging to Delineate the Basolateral Amygdala

**DOI:** 10.3390/biomedicines11020535

**Published:** 2023-02-13

**Authors:** Andre Obenaus, Eli Kinney-Lang, Amandine Jullienne, Elizabeth Haddad, Kara M. Wendel, A. Duke Shereen, Ana Solodkin, Jeffrey F. Dunn, Tallie Z. Baram

**Affiliations:** 1Department of Pediatrics, Loma Linda University School of Medicine, Loma Linda, CA 92350, USA; 2Department of Pediatrics, University of California, Irvine, CA 92697, USA; 3Department of Anatomy/Neurobiology, University of California, Irvine, CA 92697, USA; 4Department of Neurology, University of California, Irvine, CA 92697, USA; 5Department of Radiology, Hotchkiss Brain Institute, Cumming School of Medicine, University of Calgary, Alberta T2N 4N1, Canada

**Keywords:** gadolinium, magnetic resonance imaging, rodent, volumes, high field MRI, tractography, histology

## Abstract

The amygdaloid complex, including the basolateral nucleus (BLA), contributes crucially to emotional and cognitive brain functions, and is a major target of research in both humans and rodents. However, delineating structural amygdala plasticity in both normal and disease-related contexts using neuroimaging has been hampered by the difficulty of unequivocally identifying the boundaries of the BLA. This challenge is a result of the poor contrast between BLA and the surrounding gray matter, including other amygdala nuclei. Here, we describe a novel diffusion tensor imaging (DTI) approach to enhance contrast, enabling the optimal identification of BLA in the rodent brain from magnetic resonance (MR) images. We employed this methodology together with a slice-shifting approach to accurately measure BLA volumes. We then validated the results by direct comparison to both histological and cellular-identity (parvalbumin)-based conventional techniques for defining BLA in the same brains used for MRI. We also confirmed BLA connectivity targets using DTI-based tractography. The novel approach enables the accurate and reliable delineation of BLA. Because this nucleus is involved in and changed by developmental, degenerative and adaptive processes, the instruments provided here should be highly useful to a broad range of neuroimaging studies. Finally, the principles used here are readily applicable to numerous brain regions and across species.

## 1. Introduction

The mammalian amygdala plays critical roles in emotional processing, fear, motivation and attention, along with learning and memory. The amygdala itself is a constellation of nuclei and cell types that are remarkably conserved among mammalian species, including rat, monkey and human [1]. The amygdala is primarily composed of the basolateral (BLA) amygdala (lateral, basal, and accessory basal nuclei), and the central, medial and cortical nuclei, which have been termed the amygdaloid complex [2,3]. Among these nuclei, the BLA is thought to be integral to the processing of affective memories [4,5]. In humans, connectivity studies using functional magnetic resonance imaging (fMRI) and intracranial recording studies have demonstrated that the BLA complex is integral to selective memory formation and retrieval, and affective processing [6]. Similar findings in the rodent have shown its importance in associative learning and fear processing [7,8]. 

The amygdala figures prominently in several human disease states, including temporal lobe epilepsy and several types of dementia, as well as in major psychiatric disorders. Indeed, changes in amygdala volume have been reported in depression and stress [9,10]. We have previously demonstrated the important role of the amygdala (and the BLA) in early life adversity/stress [11,12]. The importance of the amygdala, and particularly the BLA, in normal brain function and in neuropsychiatric diseases, demands the establishment of imaging tools that can clearly delineate its anatomical boundaries, but they are currently lacking.

Anatomical studies of the rodent amygdala have provided details about its cellular composition, connectivity, and functional roles [13,14,15,16,17,18]. Human anatomical studies are few, and neuroimaging approaches have yielded a wealth of information. These neuroimaging studies range from volumetric assessments [19,20,21] to functional MRI studies [22,23]. In many of these studies, both the volume and the location of the amygdala are based on atlas templates that provide general information, but do not provide details about the specific nuclei within the amygdala. Loss of detail is often the norm in imaging studies due to large slice thickness; however, the lack of gray matter contrast in the temporal lobe is the primary cause underlying the paucity of studies of the BLA. Recently, attempts have been made to resolve these issues using diffusion tensor imaging (DTI) approaches [24,25,26].

Like the human brain, traditional neuroimaging (T1, T2) of the rodent amygdala has yielded poor differentiation between the amygdala and the surrounding structures, such as the piriform cortex and the caudate putamen. Even more problematic is the differentiation of the BLA from other amygdala nuclei, such as the central and medial portions of the amygdala. Several strategies have been applied to enhance contrast and enable anatomy and functional connectivity, including manganese-enhanced imaging (MEMRI) [27,28]. Notably, these approaches have been largely unsuccessful in parcellating the specific amygdala nuclei, including the BLA. Others have utilized the general delineation of the amygdala, again without distinction of its component nuclei [29]. 

Diffusion MRI, including DTI, capitalizes on the Brownian motion of water molecules in the brain (and other tissues of interest), where the tissue microstructure (white versus gray matter) preferentially limits the diffusion of water. For example, in white matter, water will preferentially move along axons, but perpendicular diffusion is limited by myelin and other such boundaries. This anisotropic water diffusion can be mapped using a minimum of six independent diffusion measurements to generate a directional tensor that provides insights into the directionality of water movement. Parametric maps can be generated for axial, radial and mean diffusivity, and fractional anisotropy that describe tissue features that can be exploited for insights into normal brain function and how these metrics are altered in disease. This directionality from DTI images can be further exploited to visualize connections between brain regions using tractography. For additional details about DTI theory and its applications, the reader is referred to these selected publications [30,31,32]. DTI and its variants have been utilized to identify and visualize alterations, predominately in white matter in both rodents and humans [33,34]. The use of DTI to probe grey matter microstructure is lacking, particularly in rodent studies, but is starting to gain momentum. Finally, while rodent atlases are emerging with excellent definition of amygdalar boundaries [35], many lack subdivisions, including the BLA and related nuclei. Thus, there is a need for identifying a reliable and robust method for visualizing the individual nuclei of the amygdala and particularly the BLA. 

We hypothesized that conventional imaging would not provide adequate delineation of the BLA, but DTI and its ability to report microstructure could provide enhanced visual and quantitative assessments of the BLA. Thus, we report here on a novel strategy using high-resolution DTI combined with a slice-shifting approach to visualize and extract BLA volumes in the adult male rat. The BLA boundaries identified using MRI-derived data were in excellent concordance with independent histological and immunohistochemical measures derived from the same brains. We further confirmed the accuracy of these BLA regions using DTI tractography. Finally, we examined reproducibility with inter- and intra-rater assessments. Our results provide an MR imaging strategy to accurately identify and quantify the BLA, which can also be applicable to additional brain regions, as well as across species. 

## 2. Materials and Methods

### 2.1. Animals

All protocols were approved by the Loma Linda University Animal Health and Safety Committee and are in compliance with federal regulations (protocol #813033, approved 09/17/2013). All experiments were conducted following the ARRIVE guidelines. Male adult Sprague Dawley rats (329.8 ± 3.3 g, Harlan; n = 24) recovered in the vivarium for 5–7 days prior to transcardiac perfusion using 4% paraformaldehyde (PFA, Electron Microscopy Sciences, Hatfield, PA, USA). Though imaging of either the brain in the cranial vault [30] or of the brain alone has been reported, brain-only samples have been used to generate atlases [31]. Therefore, we elected to remove the brains from the cranial vault. Brains were postfixed in 4% PFA, washed and stored at 4 °C in 0.1M PBS/0.05% sodium azide until imaging and histology. Together, these procedures reduced the potential for artifacts particularly at high field strengths. Prior to imaging, brains were placed in Fluorinert (FC-770, SynQuest Labs, Inc., Alachua, FL, USA) to facilitate susceptibility matched imaging.

### 2.2. T1- and T2-Weighted Magnetic Resonance Imaging (MRI)

Ex vivo brains underwent high resolution T2-weighted anatomical MRI using a 11.7 T Bruker Avance scanner (Bruker Biospin, Billerica, MA, USA). The multiecho sequence had the following acquisition parameters: a 256 × 256 matrix, 25 slices covering the whole brain at 0.6 mm slice thickness, 2 cm field of view, repetition time/echo = 2903/10.2 ms, 10 echoes, two averages for a scan time of 25 min. This resulted in a 78 × 78 × 600 µm/voxel resolution. 

A subset of animals (n = 6) underwent ultra-high resolution T1-weighted anatomical MRI using the 11.7 T Bruker Avance scanner. The sequence used was a 3D Rapid Acquisition with relaxation enhancement (3D RARE), with a 256^3^ matrix, 2 cm field of view and 78 µm slice thickness, repetition time/echo = 2388/15 ms and a single average (total scan time = ~5 h). This resulted in a 78 × 78 × 78 µm/voxel isotropic resolution. 

An additional subset of animals underwent T2- and T1-weighted imaging as described above but after post-fixative incubation with Gd-DTPA (1.5%, Magnevist^®^, Bayer, Reading, UK) (n = 3) or with Gd-DTPA (1.5%) mixed in with the PFA (n = 3) and followed post-fixation incubation for 3–5 days. Both groups were imaged using the same scan parameters. A control group without contrast agent (n = 3) was also imaged in parallel.

### 2.3. Diffusion Tensor Imaging (DTI) 

High-resolution DTI was performed on the same rat brains as reported for standard MRI (see above). DTI-MR images were acquired using a 9.4 T Bruker Biospin MR imaging system (Paravision v.5.1, RRID:SCR_001964). Ex vivo brains were carefully positioned inside a 5 mL plastic syringe and submerged in Fluorinert to eliminate any background noise and to increase the signal-to-noise ratio (SNR). Each DTI acquisition consisted of 50 slices, 0.5 mm thick encompassing the entire brain, a 1.92 × 1.92 cm field of view with a data matrix size of 128 × 128, which was zero-filled to 256 × 256 matrix during image reconstruction. A four-shot echo-planar imaging (EPI) sequence was used to acquire four averages of diffusion-weighted images with b values of 0 (5 images) and 3000 s/mm^2^ (30 images in non-collinear directions), a diffusion pulse width of 4 ms and interpulse duration of 20 ms with a repetition time of 12,500 ms and an echo time of 36 ms for an acquisition time of ~2hr. DTI data were post-processed using DSI Studio (National Taiwan University: http://dsi-studio.labsolver.org, RRID:SCR_009557). Raw Bruker data were imported into DSI Studio, where fractional anisotropy (FA), axial diffusivity (AD), radial diffusivity (RD) and trace (ADC) parametric maps were generated. In addition, primary, secondary, and tertiary diffusion eigenvalue (λ1, λ2, and λ3) maps were calculated using FSL (http://fsl.fmrib.ox.ac.uk/fsl/, accessed on 1 June 2020 RRID:SCR_002823).

For tractography, the BLA was outlined on a single high contrast-to-noise ratio (CNR) direction (direction 11, Bregma: −3.14 mm). Deterministic tractography was then performed using the following global parameters: angular threshold = 60; step size = 0.05 mm; smoothing = 0.60. The fiber threshold was optimized by DSI Studio to maximize the variance between the background and foreground and between subjects. One million unrestricted seeds were placed for the entire brain, and the tracts passing through the BLA were extracted.

Directionally specific DTI data were acquired at 11.7 T to accurately determine amygdala volumes (see below). CNR were calculated using the following equation:CNR = (siBLA − siCTX)/stdevNOISE, 
where siBLA is the signal intensity in the BLA, siCTX is the signal intensity of the cortex directly above the hippocampus, and stdevNOISE is the standard deviation of the noise in a region of interest (ROI) outside the brain tissue. CNR ROIs were inserted from a template and only the BLA ROI was modified as needed to encompass its anatomical boundaries. 

To obtain enhanced volumetric accuracy whilst preserving SNR and CNR, we used a slice-shifting method, using only a single directionally encoded DTI acquisition (direction 11). We used our standard DTI coronal slice thickness of 600 µm, but after the initial acquisition, we then shifted the entire slice packet by 200 µm in the axial direction. This shift was then repeated one more time. The standard DTI imaging sequence (see above) typically yielded 2–3 600 µm thick slices through the BLA. With the slice-shifting method, we obtained 6–9 slices encompassing the BLA with an effective inter-slice distance of 200 µm. The slices from each data set (3 data sets) were combined into a single file, where each slice was anatomically contiguous using a custom MATLAB (RRID:SCR_001622) routine. This approach provided improved volumetric resolution of the BLA. 

It is generally acknowledged that increasing the field strength and associated improvements in hardware can potentially improve anatomical visualization. To examine if our findings were improved at a higher magnetic field, we imaged several samples at the Advanced Magnetic Resonance Imaging and Spectroscopy (AMRIS) facility at the University of Florida. Ex vivo MR images were acquired on an Avance III Bruker 750 MHz (17.6 T) using an optimized 2D diffusion weighted spin echo sequence, with TR = 5 s, echo time TE = 28 ms, pulse duration (δ) = 4 ms, pulse spacing (Δ) = 12 ms, and 2 averages. The in-plane FOV was 16 × 13 × 9.75 mm, with a 128 × 104 × 78 matrix, resulting in an isotropic resolution of 125 μm^3^. A single shell diffusion weighting scheme with a total of 52 diffusion directions was used, including b = 0 and b = 3000 s/mm^2^ volumes. The tractography of the 17.6T scans utilized the Waxholm atlas, which was registered to each individual animal’s space, and regional labels were applied. DSI studio and Waxholm regions of the BLA and medial prefrontal cortex (PFC) were then used to generate tractography. Deterministic tractography was performed using the following global parameters: angular threshold = 55; step size = 0.01 mm; smoothing = 0.60. One million unrestricted seeds were placed within the BLA, and tracts passing through the BLA and mPFC were extracted for visualization.

### 2.4. BLA Volumetric Analysis from DTI

We defined the BLA as composed of basal, lateral, and accessory basal nuclei of the amygdala, as previously reported [7]. Manual ROIs were carefully drawn on the right and left BLA regions on slices encompassing the amygdala on DTI parametric maps based on known anatomical locations from atlases [32]. The amygdala complex starts at approximately Bregma −1.30 and extends to −4.80 mm. The basal and lateral nuclei (BLA) are found within this antero-posterior range from −1.40 to −3.80 mm. For our analysis, the BLA was bounded by the external capsule, a white matter tract that is readily discernible on MRI. The dorsal extent of the BLA starts at the level of the rhinal fissure and extends 1.5 to 2 mm ventrally. The BLA is also bounded by the striatal, cortical (piriform) and ventricular structures (central and medial nuclei of the amygdala) in the medial aspect. Volumetric analyses of the total brain and BLA were performed on MR coronal slices using Cheshire image processing software (Hayden Image/Processing Group, Waltham, MA, USA, RRID:SCR_018225). For the volumetric data, bilateral BLA boundaries were manually drawn on each data set from the slice-shifted directionally encoded DTI image series using the boundaries as described above. Areas from each slice were extracted and multiplied by the effective interslice distance (200 µm) and slice number to obtain BLA volumes. All data were extracted and summarized in Excel. 

To further demonstrate the robustness of the described approach, we undertook inter-rater calculations of the amygdala volumes from a random subset of three animals. Rater A derived the original delineations reported herein, Rater B had limited experience with MRI or amygdala anatomy, whilst Rater C has extensive knowledge of both. Raters B and C were provided with a protocol and a coded dataset with no additional training. After initial assessments, Rater B (least experienced) was provided with additional training and knowledge of the amygdala and then retraced the volumes 60 days later. We also tested for test–retest reliability, where Rater B re-acquired all volumes 30 days after the first set of delineations. No significant differences between amygdala volumes were found (see Appendix A).

### 2.5. Histology and Immunohistochemistry

After MR imaging was completed, brains were cryoprotected in 30% sucrose solution for 12 h and frozen on dry ice. Coronal sections (30 µm) were cut and collected on Superfrost Plus (Fisher Scientific, Pittsburg, PA, USA) microscope slides (4 sections) and as free-floating sections stored in cryoprotectant (4 sections). This sequence was repeated every 120 µm. Tissue sections were placed on slides then processed for Cresyl violet (0.1%) staining as previously described [33]. Free-floating sections were processed for parvalbumin (PV) immunohistochemistry using a modified protocol [7]. Briefly, sections were removed from cryoprotectant solution and rinsed by 3 washes in PBS. Sections were then treated for 5 min with 3% H_2_O_2_ followed by non-specific sites blocking using 10% goat serum and 0.2% Triton X for 1.5h. Sections were incubated overnight with a PV antibody (1:200, #NB120−11427, Novus Biologicals, Littleton, CO, USA, RRID:AB_791498) at 4 °C in the blocking buffer. After 3 rinses in PBS, sections were incubated with biotinylated goat anti-rabbit IgG antibody (1:500, #BA-1000, Vector laboratories, Burlingame, CA, USA, RRID:AB_2313606) in PBS with 5% goat serum and 0.01% Triton X for 1 h at room temperature. Following another 3 washes in PBS, sections were incubated in the avidin–biotin–peroxidase complex solution (Vector laboratories) for 30 min. Sections were then incubated in 0.012% 3,3′ diaminobenzidine solution containing 0.01% H_2_O_2_ to produce a brown reaction product. Sections were mounted on slides and dehydrated prior to being cover-slipped using Permount mounting medium (Fisher Scientific). For the MBP (myelin basic protein) staining, free-floating sections were washed, incubated in blocking buffer (2% goat serum in PBS) for 1.5 h, then incubated overnight with anti-rat MBP antibody (1:250, MAB386, Millipore, Billerica, MA, USA, RRID:AB_94975) in PBS with 0.5% bovine serum albumin. Alexa fluor 488 goat anti-rat IgG secondary antibody (1:1000, A-11006, Life Technologies, Carlsbad, CA, USA, RRID:AB_2534074) was used for fluorescent detection of MBP.

Histological and immunohistochemical stained slides were imaged using a Keyence BZ-X700 microscope (Keyence Corp., Osaka, Japan) capturing the entire section at magnification 10X which was reconstructed using the XY-stitching feature. Total brain and BLA areas were manually delimited using anatomically defined boundaries (PV staining, internal capsule laterally, central nucleus medially and extending up ventrally to ~1.2 mm from the base of the cortical tissues) and data were extracted from each section using BZ-X analyzer software (version 1.2.0.1, RRID:SCR_017375) with the exact same ROI protocol as used for the MRI data. Histological BLA data were obtained by an investigator who did not undertake the MRI analysis.

### 2.6. Statistical Analysis

All measurements and analyses were performed without knowledge of groups. Histological and MRI data for BLA volumes (total, left and right) were tested for distribution normality using Shapiro–Wilk test, and all data groups were normally distributed (*p* values ranged from 0.31 to 0.99). One-way analysis of variance (ANOVA), and Student’s t-tests were performed using GraphPad Prism 5.0 (GraphPad, San Diego, CA, USA, RRID:SCR_002798), followed by Bonferroni’s post hoc tests when appropriate. Correlation analyses utilized were performed using matching data and were corrected using a Geisser–Greenhouse correction. Data are presented as mean ± SEM.

## 3. Results

### 3.1. Neuroimaging and Contrast Enhancement of the Rodent Amygdala

Given that the adult rat brain volume is ~2 cc, we undertook resolution enhancement using high-field MRI. We eliminated motion artifacts and increased both CNR and SNR by using ex vivo imaging. Both approaches aimed to improve visualization of the amygdala and its component nuclei. As noted in Figure 1a,b, conventional imaging (T2WI) resulted in ambiguous boundaries of the amygdala and specifically the BLA. For example, standard 2D T2-weighted imaging (78 µm in plane with 600 µm thick slices; n = 3) provided reasonable visual contrast of many brain structures, including white matter tracts and the hippocampal formation; however, contrast in the temporal regions including the amygdaloid complex remained poor.

### 3.2. Exogenous Contrast Does Not Enhance Definition of the BLA 

Others have enhanced contrast among brain structures using exogenous contrast agents, including gadolinium-based compounds [36]. Accordingly, we employed two distinct approaches using Gd-DTPA, perfusion and incubation, to increase contrast in the amygdala and visualize its nuclei (Figure 2). Although there was enhancement of signal-to-noise ratios within the amygdala and the temporal lobe (PFA: 35.0, PFA, Gd incubation: 43.7, PFA + Gd, Gd incubation: 40.7), this strategy did not directly improve visualization of amygdala nuclei. The method did enhance contrast in hippocampus and its sub-layers, as reported by others [27,28,36]. 

### 3.3. High Resolution Neuroimaging and Conventional DTI Do Not Enhance BLA Definition

Since standard neuroimaging, either with or without contrast, did not appreciably improve discrimination of the rodent amygdala and its nuclei from the surrounding gray matter, we opted to amplify the MR resolution. However, high resolution isotropic volumetric acquisitions (3DRARE; 78 µm isotropic, 5 h acquisition) did not provide any additional visual improvements in contrast to the BLA (Figure 1a,b). Therefore, we turned to DTI as an alternative method for enhancing contrast [34]. We acquired DTI data using a scheme encompassing 30 directional vectors (78 µm in plane, 600 µm thick slices) (n = 3). However, the anatomical boundaries of the amygdala were not appreciably enhanced visually nor better delineated on the DTI derived parametric maps, including the resultant FA, trace or eigenvalue images, in comparison to routine T1 or T2 imaging (Figure 1a,b).

### 3.4. Specific Directionally Encoded DTI Directions Enhance BLA Visualization

Whereas the DTI aggregate parametric maps did not improve visualization of the BLA, we reasoned that specific directional encoding vectors might yield enhanced CNR between BLA and surrounding structures. Such an approach has not been previously reported. To test this hypothesis, we used known anatomical landmarks and the enhanced CNR of the DTI images to draw the BLA ROIs on a single slice (Bregma = −3.30 mm) on each of the directionally encoded DTI images, and the CNR was calculated (Figure 3 and Figure 4). An example of all the 30 directional DTI images is shown in Appendix A. 

Visual examination of directionally encoded images at the level of the BLA revealed that the structure could be more readily observed in some directions compared to others (Figure 3a). Indeed, in some images, the enhanced visualization was only unilateral, and in select directions, the BLA could be easily observed bilaterally. In contrast, many directions had poor or no observable BLA boundaries. 

Analyzing CNR independently for each of the individual directional DTI vectors, we identified several vectors that yielded CNR signals that were greater than 4. Specifically, directions 2 and 11 had CNR of 4.93 and 6.47, respectively, which provided optimal contrast simultaneously in both the right and left BLA (Figure 3b, Appendix A). Interestingly, we also observed vectors that provided increased CNR (4.08 to 8.10) for either the right or the left BLA, (right-direction 2, 3, 8, 11, 19, 21, 24; left direction 2, 5, 20, 25, 26, 28, 29). This finding of unequal contrast between the right and left BLA is expected, due to the different intrinsic properties of the brain structures lying either medial or lateral to the BLA. In addition, the non-homogenous cellular composition of the BLA itself likely contributes to diffusion differences along its left and right borders. Considering each side separately, CNR in the side-specific directions ranged from 4.08 to 8.10, a range that enhanced visualization. Only directional vector images that contained simultaneous increased CNR in both BLA regions were utilized for volumetric analysis. 

To further confirm our CNR approach for identifying optimal DTI vectors, we undertook additional CNR measures of the adjacent striatal region. We observed increased CNR (2.40 to 5.01) in directions similar to those seen in the cortical CNR results. The increased CNR exhibited 80% concordance in the right hemisphere (right-direction 3, 8, 19, 21, 24) and 83% in the left hemisphere (left direction 5, 20, 22, 26, 28, 29). The decreased CNR values from the striatum are expected due to its increased iron content [37]. 

We then tested if the approach described above could facilitate volumetric analysis of the BLA. Plasticity and volume changes of BLA, for example, as a result of augmented dendritic branching, were described using postmortem methodologies [38]. However, assessment of the BLA volume using imaging methods was hampered by the MRI slice thicknesses (often 1 mm) required for sufficient SNR within the context of a reasonable DTI acquisition timeframe. Notably, the adult rat amygdala is only ~1000 µm antero-posterior, so a slice thickness of 1 mm would theoretically yield only one or two slices, thus resulting in insufficient resolution for accurate volumes. 

### 3.5. Use of DTI and Slice-Shifting for Accurate Measures of BLA Volume

We combined DTI, and specifically acquired data utilizing the directional encoding vector 11 (Figure 4; n = 9) and sections that were acquired at 600 µm thickness. We then shifted the acquisition by 200 µm, then repeated this shift once more. This resulted in an effective slice thickness of 200 µm (Figure 4a, Appendix A). This approach yielded 6–9 slices (Figure 4a, right panel) that contained the BLA, in contrast to the 2–3 slices (Figure 4a, left panel) obtained using routine 600 µm thick section acquisitions. From these slice-shifted data we calculated a BLA volume of 1.44 ± 0.02 and 1.47 ± 0.04 mm^3^ (mean ± SEM, right and left BLA, respectively) (Figure 4 and Figure 6). The MRI-based volumes of the right and left BLA volumes were not significantly different from each other (*p* = 0.26, paired t-test) (Figure 4c). Combined BLA volumes (right and left combined) were 2.94 ± 0.05 mm^3^.

### 3.6. Histological Comparison and Validation of MRI-Derived BLA Volumes

To examine the accuracy and utility of the MRI-derived delineation and volume measurements of the BLA, we undertook direct comparison of the DTI method to two independent measurements performed directly on the same brains used for imaging. Representative histological samples from the same animal are shown in Figure 5. Cresyl violet staining of tissue sections encompassing the BLA resulted in volumes for right and left BLA of 1.36 ± 0.04 and 1.40 ± 0.06 mm^3^, respectively, and a combined volume of 2.76 ± 0.06 mm^3^ (Figure 5 and Figure 6a–c). The BLA contains large numbers of PV expressing cells, and the presence of these neurons was used to delineate the boundaries of this nucleus [8]. Employing PV immunohistochemistry resulted in a calculated BLA volume for right and left (1.45 ± 0.07, 1.41 ± 0.10 mm^3^) and combined volume of 2.87 ± 0.14 mm^3^ (Figure 5 and Figure 6a,b,d).

We then statistically examined the results of the three methods of measuring BLA volume, focusing on the comparison of the MRI-derived technique to the two direct tissue approaches. All methods yielded essentially similar values. Specifically, Cresyl violet and PV volumes were comparable (*p* = 0.60, paired t-test) for each side separately and for the combined BLA volume (*p* = 0.060, paired *t*-test) (Figure 6c–e). BLA volumes derived from the two histological methods did not differ appreciably from those derived from DTI / slice-shifting (*p* = 0.066, repeated measures one-way ANOVA). This was true also when each side was analyzed separately (right *p* = 0.20; left *p* = 0.19; repeated measures one-way ANOVA). The volumes obtained using the three methods were not overtly different, not only when the means of volumes derived from each method were compared, but also when the comparison was among BLA volumes of an individual brain that were examined using MRI or each of the two histological methods (Figure 6c,d). Interestingly, the variance among BLA volumes in the group of animals was smallest using MRI-derived analyses (Figure 6e), whereas PV delineation yielded the highest variance. The PV variance could be due in part to the rigors of the immunohistochemistry staining process on tissues. 

Using our methodology, we examined the inter-rater ability to segment the amygdala to obtain volumes from DTI. We observed close concordance between the individual who devised this method (Rater A) and a similarly experienced (MRI/amygdala) person (Rater C; −6.6% compared to Rater A) (Figure 6f, Appendix A). However, the naïve Rater (B, least experienced) over-estimated the total amygdala volumes by 50.6%. We then performed an in-depth training session for Rater B, leading to only a 14.6% difference. After retraining, there were no significant differences in the amygdala volumes between raters, whereas prior to training, Rater B volumes were significantly larger than those of A (*p* < 0.0001; one-way ANOVA). This would suggest that even a single training session with inexperienced MRI/amygdala individuals greatly improves the measurement outcomes.

### 3.7. Additional Validation of MRI-Derived BLA Regions

To further validate our DTI derived approach for the BLA, we performed two additional confirmatory measures. Firstly, we undertook immunohistochemical staining to elucidate the white matter tracts that are known to bound the BLA (Figure 7). Specifically, we used myelin basic protein (MBP) immunohistochemistry to identify the external capsule (EC) and the amygdalar capsule (AC). The BLA region we identified on high CNR DTI directions (e.g., direction 11) clearly match those seen and delineated by the white matter on the MBP-stained sections. This clear distinction was not evident in DTI low CNR directions (Figure 7).

Secondly, we undertook DTI tractography to validate the accuracy of our BLA boundaries. Whole brain tractography was performed followed by examining only those tracts that passed through the BLA based on our optimal CNR analysis (see Figure 3). The resultant streamlines revealed a compact bundle of tracts that passed anteriorly from the olfactory bulbs through the BLA and on posteriorly to the ventral hippocampus with tracts looping anteriorly to the medial prefrontal cortex (mPFC) (Figure 8a, left panel). A similar protocol was utilized on data from 17.6T acquisitions, and the further resolution of the BLA to mPFC tracts is apparent (Figure 8a, right panel). To further validate the accuracy of the BLA delineation, we dilated the entire region of interest uniformly by 2 pixels, which led to a dramatic increase in the number of unilateral tracts and the appearance of tracts crossing the midline (Figure 8b). A similar increase in tract density occurred when the original BLA regional designation was shifted leftward by 300 µm (4 pixels) (Figure 8c). The dramatic increase is due to the incorporation of the white matter encompassing the amygdalar capsule. A rightward shift of the BLA region by 300 µm also resulted in an increase in tracts, albeit being less robust (Figure 8d) and encompassing the smaller white matter tract, the external capsule. 

## 4. Discussion

The principal findings of the current study are as follows: (1) high resolution DTI, employing specific, directionally encoded vectors, enhances CNR and enables visualization of the BLA; (2) contrast-to-noise optimization for the right and left amygdala can be accomplished independently; (3) coupled with slice-shifting, the method described here yields high concordance with histological BLA segmentation, allowing for accurate assessment of BLA volumes across time; and 4) tractography of BLA projections validated the delineation method. Finally, the principles described here for DTI imaging of the amygdala in the rodent are applicable to other brain regions and across species.

We show here that high-field MR diffusion imaging permits resolution and delineation of BLA boundaries in rats. This is important, as volumetric increases or decreases of this structure have been reported in a broad range of rodent models that are used extensively to study the physiological importance of the amygdala in health and disease [8,13,14,17]. Electrophysiology experiments as well as neuroanatomical studies can only be performed once, after the animal is dead (postmortem). There is a major need to provide an approach that allows for repeated and detailed evaluation of the BLA, yet there have been no anatomically accurate approaches to assess volume of the BLA from MRI. MRI allows for repeated measures that are critical to assess volumetric changes with age, after a treatment, with disease progression or as a result of other manipulations. 

To address this gap, we show here that the use of specific directionally encoded DTI results in maximal CNR that was used to enhance visualization of the BLA. Ours is the first to accurately report MRI-derived volumes of the BLA with excellent concordance to histology. A number of previous studies have attempted to derive amygdala volumes from MRI [29,39,40] using standard imaging methods. These studies reported amygdala volumes ranging from 56 to 100 mm^3^ that are not consistent with those reported from anatomical studies [1]. The principal issue appears to be insufficient anatomical information about the boundaries from which to assess amygdala volumes [24]. Even increasing the resolution by increasing the imaging times in ex vivo brains has not resulted in reliable amygdala volumes [40]. Similarly, the use of contrast agents to enhance anatomical visualization, for example, in the hippocampus, is not effective for BLA delineation (Figure 2) [36]. DTI has been used previously to enhance visualization of the human [41,42] and rodent brain [43]. However, to our knowledge, DTI has not been used to directly estimate amygdala or BLA volumes in either species. Notably, our initial approach, using DTI parametric maps (i.e., FA) did not provide enhanced visualization of the BLA nuclei. 

Solano-Castiella and colleagues reported that parametric maps derived from high field (3T) DTI could potentially subdivide the human amygdala into medial and lateral regions [26,44]. Like our results, they noted poor contrast of the amygdala on FA images. However, color-coded diffusional directions revealed sub-regions within the amygdala, but the authors noted that the relationship between these directional DTI clusters and their anatomical precision remains unexplored. These initial attempts in human studies, while important, emphasize the lack of BLA or other nuclei volumes from neuroimaging data. Interestingly, we did not observe an enhanced amygdala structure in the rodent parametric maps (FA, radial, axial and mean diffusivity). However, by examining raw directionally encoded DTI images, we found improved visualization in several of the DTI directional images. We used CNR to identify the best vector directions for BLA contrast bilaterally. We found several optimal directions based on CNR in both the right and the left BLA. Thus, features of the amygdala and its nuclei, along with their inherent anatomical connectivity and structure, could be used to parcellate specific nuclei, such as BLA. We demonstrated that DTI tractography of the BLA resulted in a compact set of tracts (streamlines) passing through the BLA regions we delineated and perturbation of the locations of the BLA region resulted in dramatic increases in the number and distribution of tracts. Further, semi-automated or fully automated methods were successfully used to segment brain regions primarily to enhance the throughput of data analysis. We and others have used a variety of computational and computer vison approaches with success [45,46,47,48], and similar approaches could be used to assist in the automated segmentation of the BLA. A variety of clustering algorithms could also be utilized to delineate the BLA independent of anatomical landmarks. 

We expanded our observation of increased CNR in the BLA on specific DTI directional vectors by using a slice-shifting technique to reliably capture the volume of the BLA whilst minimizing acquisition time but maximizing signals within the region of interest. Our MRI BLA volumes were found to be virtually identical to those from Cresyl violet staining and PV immunohistochemistry. Previous studies of BLA volumes from the histology report a wide range of volumes from 0.35 to 2.39 mm^3^ [1,49,50]. Our BLA volumes of 1.44–1.50 mm^3^ clearly fall within the range previously reported. Thus, we are confident that our novel DTI imaging strategy can be used to reliably report BLA volumes in health and disease across species. 

There is considerable interest in the BLA and in the amygdala in general, particularly within psychiatric and psychological studies. The BLA of the rodent (and of humans) is critical in encoding emotional valence, particularly in stress and anxiety [51]. We have demonstrated enhanced structural connectivity of the BLA to the prefrontal cortex in adulthood after early life stress [12]. A recent review highlights the importance of BLA activation in memory, particularly during emotional memory reactivation [52]. Amygdala volumes have also been shown to be altered in psychiatric disorders, including obsessive compulsive disorder [53], post-traumatic stress disorder [54], and in major depressive disorders [55]. However, it is important to note that the literature is inconsistent on volumetric amygdalar alterations and their importance in disease.

There are several limitations of the current study. Firstly, the histological and MRI data were not corrected for tissue shrinkage after perfusion fixation but our BLA volumes are similar to other virtually identical histological studies [1]. Further, since our MRI data were obtained from ex vivo brains after tissue fixation and we then performed cellular and immunohistochemical staining on the same brain tissue, altered tissue morphology is less of a concern. Secondly, we did not explicitly test our DTI slice-shifting approach in vivo, but we see no impediment to implementing this directly to in vivo rodents and to human data. Thirdly, in our current study, we found two directionally encoded images that provided the optimal contrast for concurrent bilateral BLA visualization. It is likely that other DTI directions on other scanners with alternate encoding schemes will observe different enhancements compared to those reported herein. We do not perceive this as a significant obstacle for clinical translation nor basic science studies since examination of the optimal contrast is required only once for each encoding scheme from individual MRI scanners. However, it is important to note that our approach to BLA identification and volumes is sensitive to the encoding direction, and the optimal directionally encoded DTI direction is likely to vary both by the number of encoding directions (i.e., 30 vs. 60) and manner in which the gradients are applied by MRI hardware/software. Thus, the optimal CNR (see Figure 3) may have to be performed for each MR scanner (at least initially) or if the DTI acquisition scheme has been altered. A fourth limitation may be the effect of head motion, which could alter our proposed in vivo applicability. In our ex vivo work herein, we have not explicitly tested motion correction schemes in the current data, but future work could evaluate the impact of motion on the sensitivity of our proposed method. Finally, in this study, manual regions of interest were performed because publicly available MRI atlases do not have nuclear parcellation of the amygdala. Manual segmentation is not optimal for large-scale studies, and future work will aim to develop suitable atlases. 

In summary, we described a novel approach for high-resolution delineation of the rodent BLA based on standard neuroimaging methods (i.e., DTI). We believe that the techniques we described can have broad implications for identification and volumetric analyses of this crucial brain structure. Increasing studies are now probing the function of the BLA (and other brain regions) as it relates to learning, memory, and neuropsychiatric diseases, where assessments of volume can provide additional insights. Moreover, once the BLA has been delineated, this “template” can and should then be utilized in other imaging modalities, including resting-state fMRI and others. Finally, while we focused on rodent BLA, we foresee the application of this schema to non-human primates and even to human MR imaging data for increased rigor and reproducibility in future investigations.

## 5. Conclusions

The importance of the amygdala in emotional and affective signal processing is well established. However, individual nuclei within the amygdala have specific processing roles and anatomical and volumetric delineation of these nuclei based on neuroimaging is becoming increasingly important. The poor MR contrast in the temporal lobe between the amygdala nuclei limits further non-invasive inquiry. While various strategies have been used to increase differentiation between these amygdala sub-structures, there has been limited success. To resolve this impasse in rodents, we utilized high-resolution directionally encoded DTI, in combination with a slice-shifting approach. We delineated the BLA amygdala nucleus and calculated its volume. We confirmed our DTI-derived volumes by comparing MRI results to two histological methods from the same animals and found excellent concordance. Additionally, connectivity between the BLA and its known target, the medial prefrontal cortex, confirmed the accuracy of the BLA delineation. Thus, we identified a novel approach using DTI directional encoded directions wherein there is increased contrast in the BLA (relative to the gray matter of the temporal lobe) combined with a slice-shifting approach to obtain reliable volumes of the BLA. This approach is readily translatable to clinical and pre-clinical studies aiming to understand the BLA role in normal brain and in neuro-psychiatric disorders. 

## Figures and Tables

**Figure 1 biomedicines-11-00535-f001:**
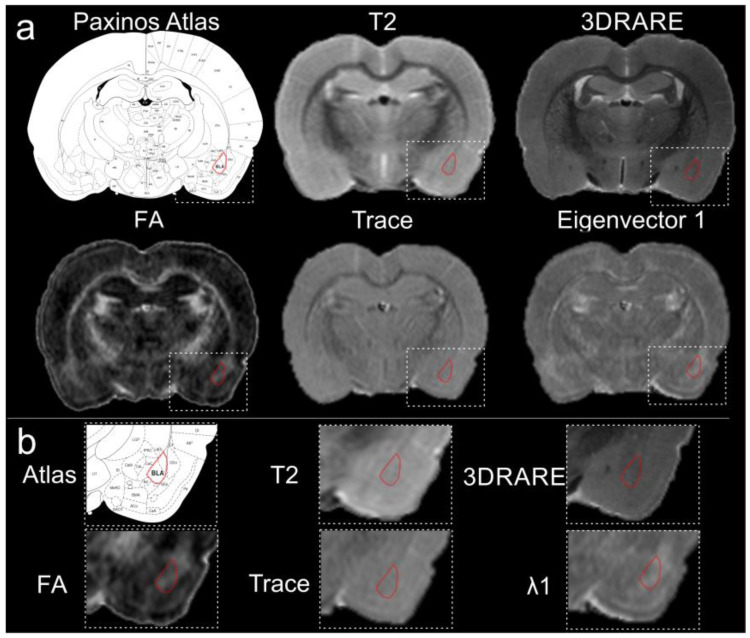
High-resolution magnetic resonance imaging of the rodent brain does not elucidate the boundaries for the basolateral amygdala. (**a**) The basolateral complex of the amygdala is highlighted (red) in several common imaging modalities including T2-weighted imaging, 3DRARE (3D rapid acquisition with relaxation enhancement) and diffusion tensor imaging parametric maps, including fractional anisotropy (FA), trace and eigenvalue λ1. The anatomical boundaries of the BLA on the MR images were created by referencing the Paxinos Atlas. (**b**) Expanded images from the MR images in (**a**), further illustrates the difficulty in identifying the anatomical boundaries of the BLA. All images were collected from the same animal at 11.7 T.

**Figure 2 biomedicines-11-00535-f002:**
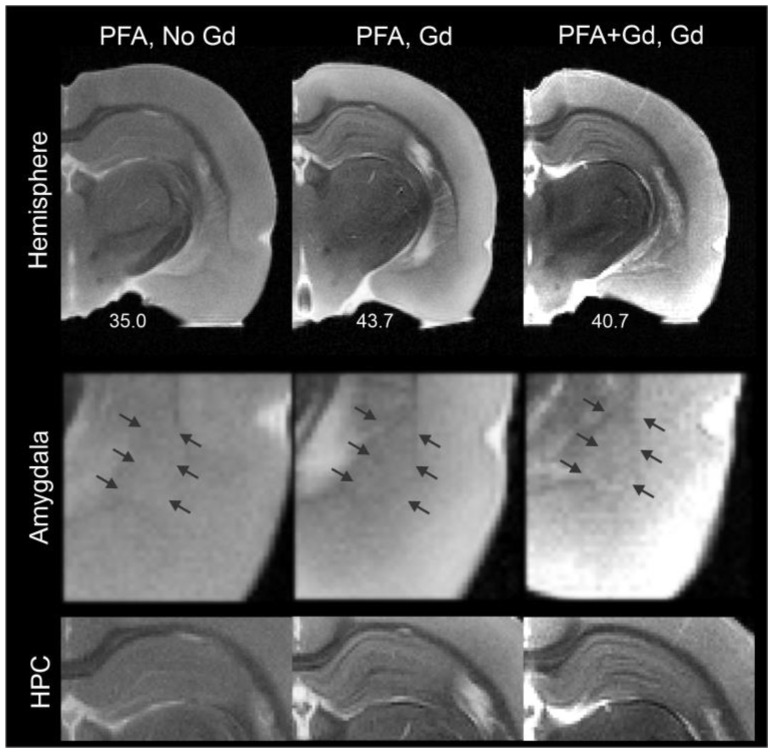
Exogenous contrast does not improve amygdala visualization. Exogenous contrast enhancement with gadolinium (Gd), either within the perfusate or by immersion of ex vivo brain tissues did not dramatically improve visualization of the amygdala or the BLA (middle panel, arrows indicate the approximate location of the BLA). In contrast and similar to other published reports, the hippocampus (HPC) exhibited enhanced contrast under both treatment paradigms (bottom panel). The SNR values were improved with Gd treatment (values are reported under the hemispheres).

**Figure 3 biomedicines-11-00535-f003:**
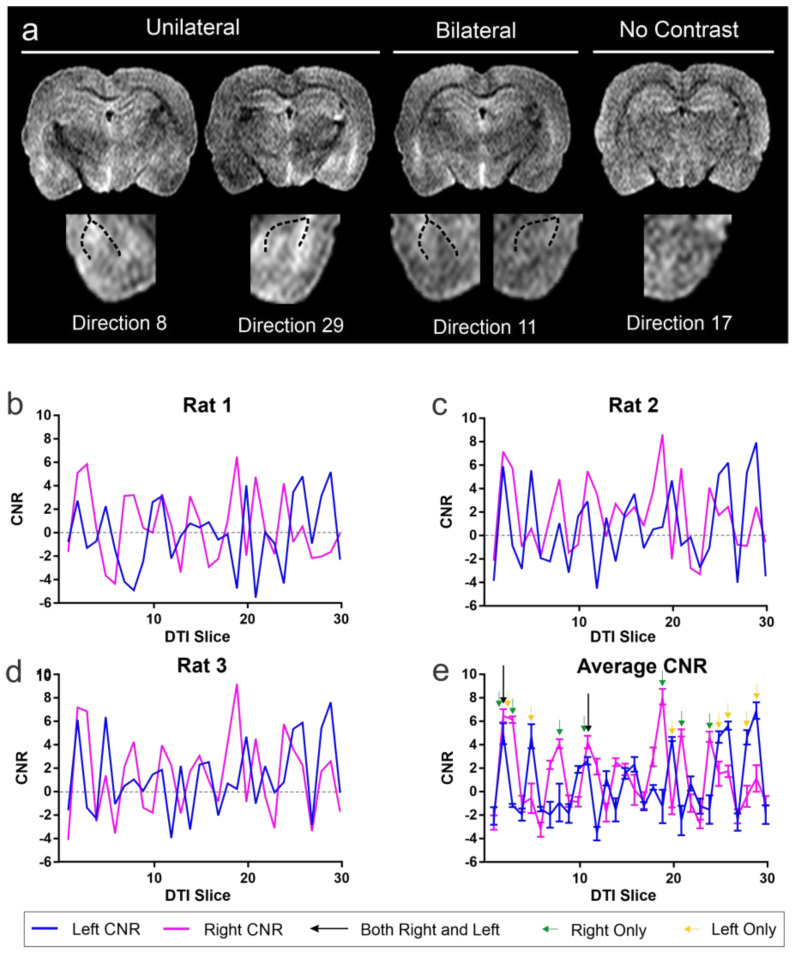
Optimal diffusion tensor directionality for identification of the basolateral complex of the amygdala. (**a**) Examples of directional diffusion tensor images wherein either left, right (unilateral) or bilateral BLA appear to have increased contrast. A tensor image in which there is no enhanced contrast within the BLA is also shown. The dotted lines demarcate the external white matter boundaries of the external capsule that bound the BLA in the temporal lobe. See also Appendix A. (**b**) Contrast-to-noise ratio (CNR) measures at the level of the BLA (Bregma, −3.30 mm) from each directional vector (see **a**) demonstrate remarkable consistency between three animals (**b–d**) in enhanced contrast in directional tensor images. (**d**) CNR measures were then averaged across all three brains to identify the optimal unilateral or bilateral CNR in the BLA. (**e**) Green arrows indicate optimal CNR in the right BLA, while yellow arrows identify the optimal directions for the left BLA. Two directions (2, 11) have optimal CNR in both BLA simultaneously. While there are optimal directions for either the left or right BLA only, there are two directional DTI images which produce the highest CNR for both the left and right BLA within the same MR image and with the least variance (direction 2 and 11). (See Appendix A).

**Figure 4 biomedicines-11-00535-f004:**
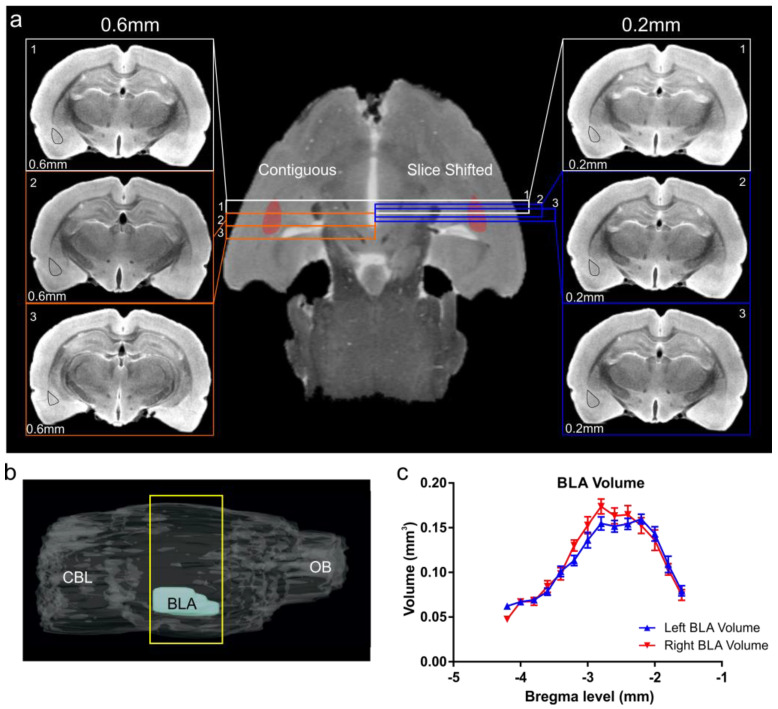
Basolateral amygdala volumes derived using a slice-shifting approach. (**a**) Conventional sequential DTI of the rat brain utilized a slice thickness (contiguous) of 0.6 mm for optimal signal to noise/contrast to noise (SNR/CNR). Thus, with this acquisition method small anatomical structures, such as the BLA, only 2–3 slices capture the volume of the BLA (left-hand panel, slice thickness 0.6 mm, isodistance between slices = 0.6 mm, left BLA region outlined in black). In contrast, a slice-shifting technique that retained the same slice thickness (for optimal SNR/CNR) but captured a series of slices through the BLA at different isodistances with offsets of 0.2 mm (right-hand panel, slice thickness 0.6 mm, isodistance between slices = 0.2 mm) provided 6–9 slices encompassing the BLA. Using slice-shifting, the small area of the basolateral complex of the amygdala is captured in a greater number of slices for accurate volumetric analysis. (**b**) Sagittal MRI view of the rat brain illustrating the region encompassed by the BLA (yellow rectangle) from which DTI data were obtained. CBL—cerebellum, OB—olfactory bulb. (**c**) The slice shifting method yields an average BLA volume in an anterior–posterior direction that is similar between right and left BLA volumes and concur with the histology and volumes reported in the literature (see Figure 5). (See Appendix A).

**Figure 5 biomedicines-11-00535-f005:**
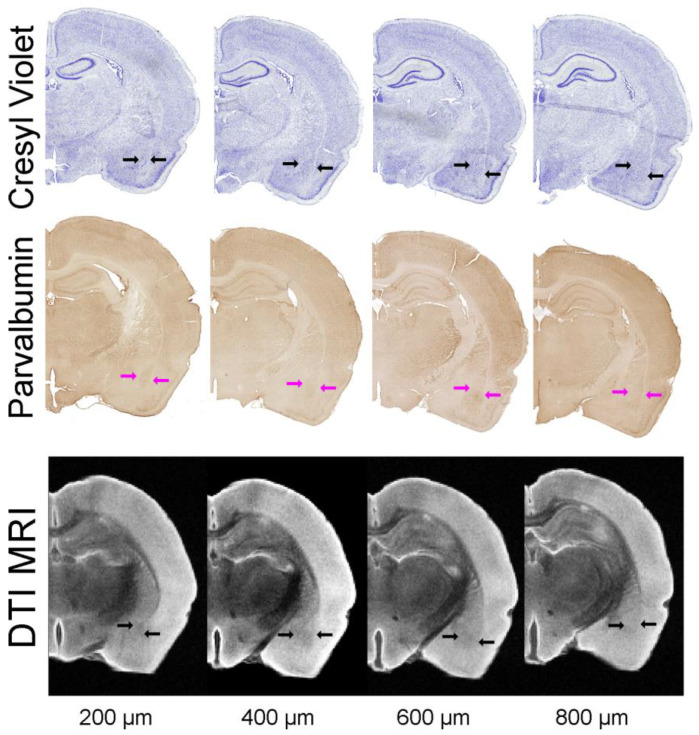
Histological delineation of the BLA. Histological sections (30 µm thick) were matched to MRI DTI slices (direction 11, 200 µm thick) for volumetric analysis. Cresyl violet staining and parvalbumin immunohistochemistry for BLA were utilized to derive BLA volumes. Blue and magenta arrows outline the BLA on the histology. The DTI eigenvalue images are from direction 11 with slices 200 µm apart. All data in this figure were derived from the same animal.

**Figure 6 biomedicines-11-00535-f006:**
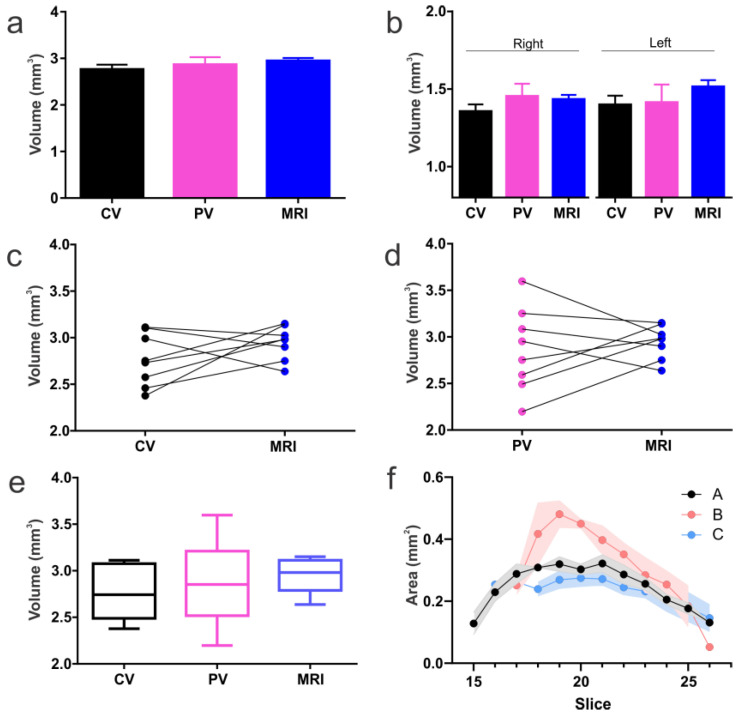
Histological and MRI derived BLA volumes. (**a**) No significant differences were found in BLA volumes extracted from Cresyl violet (CV), parvalbumin (PV) or DTI MRI. (**b**) Similarly, no significant differences were seen in volumes from either the right or left BLA. (**c**) CV anatomical data matched well to MRI. (**d**) PV data exhibited greater variability than CV in BLA volumes but also matched MRI-derived data. (**e**) Box and whisker plots illustrate that BLA volumes from MRI had the lowest variability while PV exhibited the largest. This low variability in the MRI may be due to use of all the data (6–9 slices), while histology measurements were at 120 µm intervals. (**f**) Inter-rater assessments yielded excellent concordance between the individual who devised the protocol (Rater A) and an experienced MRI/amygdala research associate (Rater C). After training, an individual (Rater B) with no MRI nor amygdala knowledge was able to reasonably delineate the BLA with a 14.6% over estimation (see Appendix A).

**Figure 7 biomedicines-11-00535-f007:**
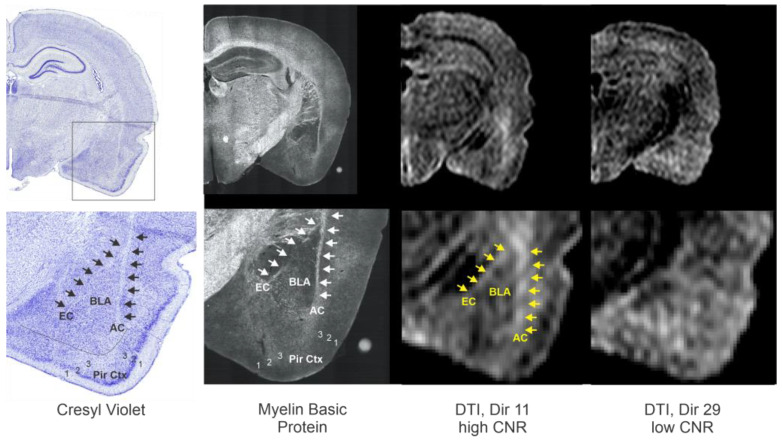
Further confirmation of the ability for DTI to delineate the BLA. To further demonstrate that a specific directional DTI could delineate the BLA, we compared Cresyl violet and myelin basic protein (MBP) stained sections to high and low CNR DTI. The MBP staining provided further verification of known white matter tracts that bound the medial and lateral borders of the BLA, the external capsule (EC) and amgdalar capsule (AC). As can be appreciated, direction 11, which exhibited high CNR values, exhibits clear boundaries of the BLA, whereas a DTI direction with low CNR has little identifiable BLA landmarks. Arrows outline the predominate white matter tracts when observed. BLA—basolateral amygdala, EC—external capsule; white matter, AC—amygdalar capsule, white matter, Pir Ctx—piriform cortex where 1,2,3 identify the cortical layers. Histology is submicron/pixel resolution, whereas DTI is 75 µm/pixel in plane and 600 µm thick. All images are from the same animal.

**Figure 8 biomedicines-11-00535-f008:**
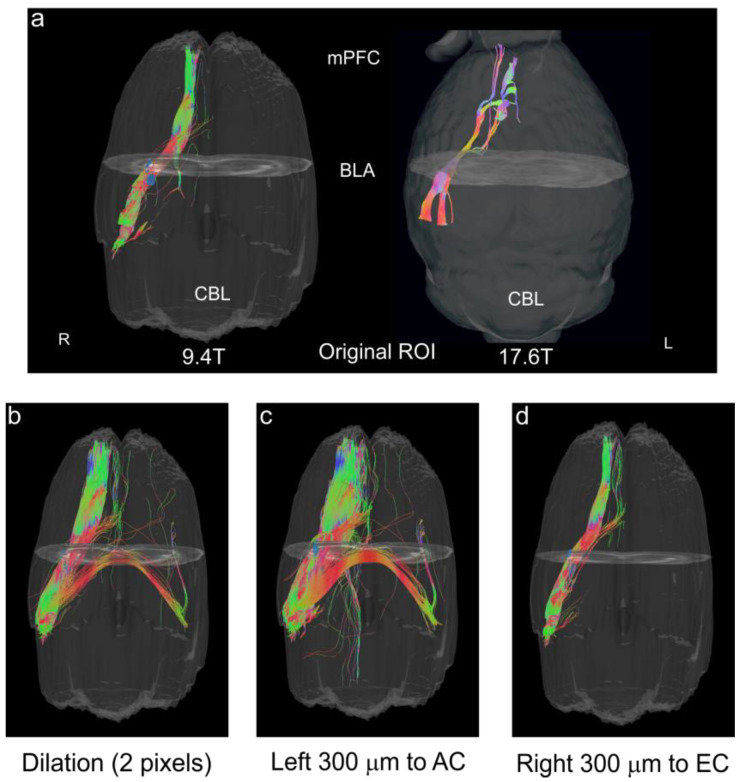
Tractography validation of the BLA regions from DTI. The BLA regions derived from high CNR directions were evaluated using tractography to evaluate known reciprocal connections between the medial prefrontal cortex (mPFC) and the BLA. (**a**) Tracts (streamlines) from the BLA regions delineated on DTI direction 8 (high right CNR; see Figure 3e), illustrate a uniform and compact connection between the mPFC and the BLA (left panel). In the right panel using an identical protocol we visualized the enhanced connections of the BLA to the medial prefrontal cortex (mPFC) at 17.6T. (**b**) Uniform dilation by 2 pixels (150 μm) of the BLA region resulted in a proliferation of tracts including bilateral projections. (**c**) A leftward shift of BLA region by 300 µm dramatically increased the number of streamlines as well as the appearance of bilateral connections. This shift included portions of the amygdalar capsule. (**d**) An identical rightward shift of 300 µm that now encompassed the smaller external capsule (EC) white matter and potentially portions of the medial amygdala also resulted in increased tract numbers.

## Data Availability

All data presented in this study are available on reasonable request from the corresponding author.

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
