# Peer review of "Seeking the Amygdala: Novel Use of Diffusion Tensor Imaging to Delineate the Basolateral Amygdala"

_biomedicines, 2023, doi:10.3390/biomedicines11020535_

Round 1

Reviewer 1 Report

The paper is interesting and, to the best of my knowledge, quite new, deserving attention for publication. Overall, it is well written and scientifically sound. I do not have major points to be addressed, but just a couple minor remarks, as stated below:

- Statistical analysis: why using a parametrical approach in this study? Any prior test to check for normality of the distribution was performed?

- Conclusions: A stronger take-home message, including a frank acknowledgment of possible future directions for the present study, is needed.

Author Response

Comments and Suggestions for Authors

The paper is interesting and, to the best of my knowledge, quite new, deserving attention for publication. Overall, it is well written and scientifically sound. I do not have major points to be addressed, but just a couple minor remarks, as stated below:

RESPONSE: We appreciate the reviewer’s interest and comments for our manuscript.

- Statistical analysis: why using a parametrical approach in this study? Any prior test to check for normality of the distribution was performed?

RESPONSE: We thank the reviewer for this question. We used a parametric approach as the data were normally distributed. We had not explicitly noted this in the manuscript and have now included a statement in the methods clarifying this point. We have also now noted in more detail the statistical test utilized whenever a p-value was reported, for additional clarity. These changes are noted in red text.

Pg5, Sec 2.6:  Histological and MRI data for BLA volumes (total, left and right) were tested for distribution normality using Shapiro-Wilk test and all data groups were normally distributed (p values ranged from 0.31 to 0.99).

- Conclusions: A stronger take-home message, including a frank acknowledgment of possible future directions for the present study, is needed.

RESPONSE: We are happy to include a stronger take-home message. We have now added a final paragraph at the end of the discussion just prior to the conclusions, that is more emphatic about our findings and future directions.

Pg 16, Sec 4, last paragraph: In summary, we have described a novel approach for high resolution delineation of the rodent BLA based on standard neuroimaging methods (i.e. DTI). We believe that the techniques we described can have broad implications for identification and volumetric analyses of this crucial brain structure. Increasing studies are now probing the function of the BLA (and other brain regions) as it relates to learning, memory, and neuropsychiatric diseases, where assessments of volume can provide additional insights. Moreover, once the BLA has been delineated, this “template” can and should then be utilized in other imaging modalities including resting state fMRI and others. Finally, while we focused on rodent BLA, we foresee the application of this schema to non-human primates and even to human MR imaging data for increased rigor and reproducibility in future investigations.

Reviewer 2 Report

Thanks for recommending me as a reviewer. In this paper, the authors described a new diffusion tensor imaging approach that enhances the contrast for optimal identification of the rodent brain BLA in magnetic resonance images. The authors used this methodology in conjunction with a slice movement method to accurately measure the BLA volume. The authors then validated the results by direct comparison with existing techniques based on histology and cell identification (parvalbumin) to define BLA in the same brain used for MRI. If the authors complete minor revisions, the quality of the study will be further improved.

1. The introduction section is well written. If the authors describe the theoretical background related to the novel use of diffusion tensor imaging to delineate the basolateral amygdala in more detail in the introductory section, it will help readers understand.

2. line 246-252: Statistical Analysis: If the authors describe statistical analysis in more detail, it can help readers understand.

3. Discussion section: The authors fully explained the limitations of the study.

Author Response

Comments and Suggestions for Authors

Thanks for recommending me as a reviewer. In this paper, the authors described a new diffusion tensor imaging approach that enhances the contrast for optimal identification of the rodent brain BLA in magnetic resonance images. The authors used this methodology in conjunction with a slice movement method to accurately measure the BLA volume. The authors then validated the results by direct comparison with existing techniques based on histology and cell identification (parvalbumin) to define BLA in the same brain used for MRI. If the authors complete minor revisions, the quality of the study will be further improved.

  1. The introduction section is well written. If the authors describe the theoretical background related to the novel use of diffusion tensor imaging to delineate the basolateral amygdala in more detail in the introductory section, it will help readers understand.

RESPONSE: We are pleased that you found the introduction well written. We are happy to include some more theory on the based of DTI to the introduction.

Pg 2, Introduction, Line 70:  Diffusion MRI, including DTI, capitalizes on the Brownian motion of water molecules in the brain (and other tissues of interest) where tissue microstructure (white versus gray matter) preferentially limits the diffusion of water. For example, in white matter, water will preferentially move along axons but perpendicular diffusion is limited by myelin and other such boundaries. This anisotropic water diffusion can be mapped using a minimum of six independent diffusion measurements to generate a directional tensor that provides insights into the directionality of water movement. Parametric maps can be generated for axial, radial and mean diffusivity and fractional anisotropy that describe tissue level features that can be exploited for insights into normal brain function and how these metrics are altered in disease. This directionality from DTI images can be further exploited to visualize connections between brain regions using tractography. For additional details about DTI theory and its applications, the reader is referred to these selected publications [30-32].

  1. line 246-252: Statistical Analysis: If the authors describe statistical analysis in more detail, it can help readers understand.

RESPONSE: We have provided more statistical detail both in the methods and in the manuscript itself wherein we have detailed the statistical test utilized. As requested by Reviewer 1 we have now included normality testing in support of our choice of statistical tests. (see above and in manuscript pg. 5).

  1. Discussion section: The authors fully explained the limitations of the study.

RESPONSE: Thank you for noting that the limitations are sufficient.